# Maximum a Posteriori Policy Optimisation

**Abbas Abdolmaleki, Jost Tobias Springenberg, Yuval Tassa, Remi Munos,**
**Nicolas Heess, Martin Riedmiller**
DeepMind, London, UK
`{aabdolmaleki,springenberg,tassa,munos,heess,riedmiller}@google.com`

## Abstract

We introduce a new algorithm for reinforcement learning called Maximum a-posteriori Policy Optimisation (MPO) based on coordinate ascent on a relative-entropy objective. We show that several existing methods can directly be related to our derivation. We develop two off-policy algorithms and demonstrate that they are competitive with the state-of-the-art in deep reinforcement learning. In particular, for continuous control, our method outperforms existing methods with respect to sample efficiency, premature convergence and robustness to hyperparameter settings.

## 1 Introduction

Model free reinforcement learning algorithms can acquire sophisticated behaviours by interacting with the environment while receiving simple rewards. Recent experiments (Mnih et al., 2015; Jaderberg et al., 2016; Heess et al., 2017) successfully combined these algorithms with powerful deep neural-network approximators while benefiting from the increase of compute capacity.

Unfortunately, the generality and flexibility of these algorithms comes at a price: They can require a large number of samples and – especially in continuous action spaces – suffer from high gradient variance. Taken together these issues can lead to unstable learning and/or slow convergence. Nonetheless, recent years have seen significant progress, with improvements to different aspects of learning algorithms including stability, data-efficiency and speed, enabling notable results on a variety of domains, including locomotion (Heess et al., 2017; Peng et al., 2016), multi-agent behaviour (Bansal et al., 2017) and classical control (Duan et al., 2016).

Two types of algorithms currently dominate scalable learning for continuous control problems: First, Trust-Region Policy Optimisation (TRPO; Schulman et al. 2015) and the derivative family of Proximal Policy Optimisation algorithms (PPO; Schulman et al. 2017b). These policy-gradient algorithms are on-policy by design, reducing gradient variance through large batches and limiting the allowed change in parameters. They are robust, applicable to high-dimensional problems, and require moderate parameter tuning, making them a popular first choice (Ho & Ermon, 2016). However, as on-policy algorithms, they suffer from poor sample efficiency.

In contrast, off-policy value-gradient algorithms such as the Deep Deterministic Policy Gradient (DDPG, Silver et al. 2014; Lillicrap et al. 2016), Stochastic Value Gradient (SVG, Heess et al. 2015), and the related Normalized Advantage Function formulation (NAF, Gu et al. 2016b) rely on experience replay and learned (action-)value functions. These algorithms exhibit much better data efficiency, approaching the regime where experiments with real robots are possible (Gu et al., 2016a; Andrychowicz et al., 2017). While also popular, these algorithms can be difficult to tune, especially for high-dimensional domains like general robot manipulation tasks.

In this paper we propose a novel off-policy algorithm that benefits from the best properties of both classes. It exhibits the scalability, robustness and hyperparameter insensitivity of on-policy algorithms, while offering the data-efficiency of off-policy, value-based methods.

To derive our algorithm, we take advantage of the duality between control and estimation by using Expectation Maximisation (EM), a powerful tool from the probabilistic estimation toolbox, in order to solve control problems. This duality can be understood as replacing the question "what are the actions which maximise future rewards?" with the question "assuming future success in maximising

rewards, what are the actions most likely to have been taken?". By using this estimation objective we have more control over the policy change in both E and M steps, yielding robust learning. We show below that several algorithms, including TRPO, can be directly related to this perspective. We leverage the fast convergence properties of EM-style coordinate ascent by alternating a non-parametric data-based E-step which re-weights state-action samples, with a supervised, parametric M-step using deep neural networks.

We evaluate our algorithm on a broad spectrum of continuous control problems including a 56 DoF humanoid body. All experiments used the same optimisation hyperparameters [1]. Our algorithm shows remarkable data efficiency often solving the tasks we consider an order of magnitude faster than the state-of-the-art. A video of some resulting behaviours can be found here **dropbox.com/s/pgcmjst7t0zwm4y/MPO.mp4**.

## 2 BACKGROUND AND NOTATION

### 2.1 RELATED WORK

Casting Reinforcement Learning (RL) as an inference problem has a long history dating back at least two decades (Dayan & Hinton, 1997). The framework presented here is inspired by a variational inference perspective on RL that has previously been utilised in multiple studies; c.f. Dayan & Hinton (1997); Neumann (2011); Deisenroth et al. (2013); Rawlik et al. (2012); Levine & Koltun (2013); Florensa et al. (2017).

Particular attention has been paid to obtaining *maximum entropy* policies as the solution to an inference problem. The penalisation of determinism can be seen encouraging both robustness and simplicity. Among these are methods that perform trajectory optimisation using either linearised dynamics (Todorov, 2008; Toussaint, 2009; Levine & Koltun, 2013) or general dynamics as in path integral control (Kappen, 2005; Theodorou et al., 2010). In contrast to these algorithms, here we do not assume the availability of a transition model and avoid on-policy optimisation. A number of other authors have considered the same perspective but in a model-free RL setting (Neumann, 2011; Peters et al., 2010a; Florensa et al., 2017; Daniel et al., 2016) or inverse RL problems (Ziebart et al., 2008). These algorithms are more directly related to our work and can be cast in the same (EM-like) alternating optimisation scheme on which we base our algorithm. However, they typically lack the maximisation (M)-step – with the prominent exception of REPS, AC-REPS, PI$^2$-GPS and MDGPS (Peters et al., 2010a; Wirth et al., 2016; Chebotar et al., 2016; Montgomery & Levine, 2016) to which our algorithm is closely related as outlined below. An interesting recent addition to these approaches is an EM-perspective on the PoWER algorithm (Roux, 2016) which uses the same iterative policy improvement employed here, but commits to parametric inference distributions and avoids an exponential reward transformation, resulting in a harder to optimise lower bound.

As an alternative to these policy gradient inspired algorithms, the class of recent algorithms for soft Q-learning (e.g. Rawlik et al. (2012); Haarnoja et al. (2017); Fox et al. (2016) parameterise and estimate a so called "soft" Q-function directly, implicitly inducing a maximum entropy policy. A perspective that can also be extended to hierarchical policies (Florensa et al., 2017), and has recently been used to establish connections between Q-learning and policy gradient methods (O'Donoghue et al., 2016; Schulman et al., 2017a). In contrast, we here rely on a parametric policy, our bound and derivation is however closely related to the definition of the soft (entropy regularised) Q-function.

A line of work, that is directly related to the "RL as inference" perspective, has focused on using information theoretic regularisers such as the entropy of the policy or the Kullback-Leibler divergence (KL) between policies to stabilise standard RL objectives. In fact, most state-of-the-art policy gradient algorithms fall into this category. For example see the entropy regularization terms used in Mnih et al. (2016) or the KL constraints employed by work on trust-region based methods (Schulman et al., 2015; 2017b; Gu et al., 2017; Wang et al., 2017). The latter methods introduce a trust region constraint, defined by the KL divergence between the new policy and the old policy, so that the expected KL divergence over state space is bounded. From the perspective of this paper these trust-region based methods can be seen as optimising a parametric E-step, as in our algorithm, but are "missing" an explicit M-step.

---

[1]With the exception of the number of samples collected between updates.

Finally, the connection between RL and inference has been invoked to motivate work on exploration. The most prominent examples for this are formed by work on Boltzmann exploration such as Kaelbling et al. (1996); Perkins & Precup (2002); Sutton (1990); O'Donoghue et al. (2017), which can be connected back to soft Q-learning (and thus to our approach) as shown in Haarnoja et al. (2017).

## 2.2 MARKOV DECISION PROCESSES

We consider the problem of finding an optimal policy $\pi$ for a discounted reinforcement learning (RL) problem; formally characterized by a Markov decision process (MDP). The MDP consists of: continuous states $s$, actions $a$, transition probabilities $p(s_{t+1}|s_t, a_t)$ – specifying the probability of transitioning from state $s_t$ to $s_{t+1}$ under action $a_t$ –, a reward function $r(s, a) \in \mathbb{R}$ as well as the discounting factor $\gamma \in [0, 1)$. The policy $\pi(a|s, \boldsymbol{\theta})$ (with parameters $\boldsymbol{\theta}$) is assumed to specify a probability distribution over action choices given any state and – together with the transition probabilities – gives rise to the stationary distribution $\mu_\pi(s)$.

Using these basic quantities we can now define the notion of a Markov sequence or trajectory $\tau_\pi = \{(s_0, a_0) \dots (s_T, a_T)\}$ sampled by following the policy $\pi$; i.e. $\tau_\pi \sim p_\pi(\tau)$ with $p_\pi(\tau) = p(s_0) \prod_{t>0} p(s_{t+1}|s_t, a_t)\pi(a_t|s_t)$; and the expected return $\mathbb{E}_{\tau_\pi}[\sum_{t=0}^\infty \gamma^t r(s_t, s_t)]$. We will use the shorthand $r_t = r(s_t, a_t)$.

## 3 MAXIMUM A POSTERIORI POLICY OPTIMISATION

Our approach is motivated by the well established connection between RL and probabilistic inference. This connection casts the reinforcement learning problem as that of inference in a particular probabilistic model. Conventional formulations of RL aim to find a trajectory that maximizes expected reward. In contrast, inference formulations start from a prior distribution over trajectories, condition a desired outcome such as achieving a goal state, and then estimate the posterior distribution over trajectories consistent with this outcome.

A finite-horizon undiscounted reward formulation can be cast as inference problem by constructing a suitable probabilistic model via a likelihood function $p(O = 1|\tau) \propto \exp(\sum_t r_t/\alpha)$, where $\alpha$ is a temperature parameter. Intuitively, $O$ can be interpreted as *the event of obtaining maximum reward by choosing an action*; or the event of succeeding at the RL task (Toussaint, 2009; Neumann, 2011). With this definition we can define the following lower bound on the likelihood of optimality for the policy $\pi$:

$$\log p_\pi(O = 1) = \log \int p_\pi(\tau)p(O = 1|\tau)d\tau \geq \int q(\tau)\Big[\log p(O = 1|\tau) + \log \frac{p_\pi(\tau)}{q(\tau)}\Big]d\tau \quad (1)$$

$$= \mathbb{E}_q\Big[\sum_t r_t/\alpha\Big] - \mathrm{KL}\Big(q(\tau)||p_\pi(\tau)\Big) = \mathcal{J}(q, \pi), \quad (2)$$

where $p_\pi$ is the trajectory distribution induced by policy $\pi(a|s)$ as described in section 2.2 and $q(\tau)$ is an auxiliary distribution over trajectories that will discussed in more detail below. The lower bound $\mathcal{J}$ is the evidence lower bound (ELBO) which plays an important role in the probabilistic modeling literature. It is worth already noting here that optimizing (2) with respect to $q$ can be seen as a regularized RL problem.

An important motivation for transforming a RL problem into an inference problem is that this allows us draw from the rich toolbox of inference methods: For instance, $\mathcal{J}$ can be optimized with the familiy of expectation maximization (EM) algorithms which alternate between improving $\mathcal{J}$ with respect to $q$ and $\pi$. In this paper we follow classical (Dayan & Hinton, 1997) and more recent works (e.g. Peters et al. 2010b; Levine & Koltun 2013; Daniel et al. 2016; Wirth et al. 2016) and cast policy search as a particular instance of this family. Our algorithm then combines properties of existing approaches in this family with properties of recent off-policy algorithms for neural networks.

The algorithm alternates between two phases which we refer to as E and M step in reference to an EM-algorithm. The E-step improves $\mathcal{J}$ with respect to $q$. Existing EM policy search approaches perform this step typically by reweighting trajectories with sample returns (Kober & Peters, 2009) or via local trajectory optimization (Levine & Koltun, 2013). We show how off-policy deep RL

techniques and value-function approximation can be used to make this step both scalable as well as data efficient. The M-step then updates the parametric policy in a supervised learning step using the reweighted state-action samples from the E-step as targets.

These choices lead to the following desirable properties: (a) low-variance estimates of the expected return via function approximation; (b) low-sample complexity of value function estimate via robust off-policy learning; (c) minimal parametric assumption about the form of the trajectory distribution in the E-step; (d) policy updates via supervised learning in the M step; (e) robust updates via hard trust-region constraints in both the E and the M step.

### 3.1 POLICY IMPROVEMENT

The derivation of our algorithm then starts from the infinite-horizon analogue of the KL-regularized expected reward objective from Equation (2). In particular, we consider variational distributions $q(\tau)$ that factor in the same way as $p_\pi$, i.e. $q(\tau) = p(s_0) \prod_{t>0} p(s_{t+1}|s_t, a_t) q(a_t|s_t)$ which yields:

$$\mathcal{J}(q, \boldsymbol{\theta}) = \mathbb{E}_q \Big[ \sum_{t=0}^{\infty} \gamma^t \big[ r_t - \alpha \mathrm{KL}\big(q(a|s_t) \| \pi(a|s_t, \boldsymbol{\theta})\big) \big] \Big] + \log p(\boldsymbol{\theta}). \tag{3}$$

Note that due to the assumption about the structure of $q(\tau)$ the KL over trajectories decomposes into a KL over the individual state-conditional action distributions. This objective has also been considered e.g. by Haarnoja et al. (2017); Schulman et al. (2017a). The additional $\log p(\boldsymbol{\theta})$ term is a prior over policy parameters and can be motivated by a maximum a-posteriori estimation problem (see appendix for more details).

We also define the regularized Q-value function associated with (3) as

$$Q_\theta^q(s, a) = r_0 + \mathbb{E}_{q(\tau), s_0=s, a_0=a} \left[ \sum_{t \geq 1}^{\infty} \gamma^t \big[ r_t - \alpha \mathrm{KL}(q_t \| \pi_t) \big] \right], \tag{4}$$

with $\mathrm{KL}\big(q_t \| \pi_t\big) = \mathrm{KL}\big(q(a|s_t)) \| \pi(a|s_t, \boldsymbol{\theta})\big)$. Note that $\mathrm{KL}\big(q_0 \| \pi_0\big)$ and $p(\theta)$ are not part of the Q-function as they are not a function of the action.

We observe that optimizing $\mathcal{J}$ with respect to $q$ is equivalent to solving an expected reward RL problem with augmented reward $\tilde{r}_t = r_t - \alpha \log \frac{q(a_t|s_t)}{\pi(a_t|s_t, \boldsymbol{\theta})}$. In this view $\pi$ represents a default policy towards which $q$ is regularized – i.e. the current best policy. The MPO algorithm treats $\pi$ as the primary object of interest. In this case $q$ serves as an auxiliary distribution that allows optimizing $\mathcal{J}$ via alternate coordinate ascent in $q$ and $\pi_{\boldsymbol{\theta}}$, analogous to the expectation-maximization algorithm in the probabilistic modelling literature. In our case, the E-step optimizes $\mathcal{J}$ with respect to $q$ while the M-step optimizes $\mathcal{J}$ with respect to $\pi$. Different optimizations in the E-step and M-step lead to different algorithms. In particular, we note that for the case where $p(\boldsymbol{\theta})$ is an uninformative prior a variant of our algorithm has a monotonic improvement guarantee as show in the Appendix A.

### 3.2 E-STEP

In the E-step of iteration $i$ we perform a partial maximization of $\mathcal{J}(q, \boldsymbol{\theta})$ with respect to $q$ given $\theta = \theta_i$. We start by setting $q = \pi_{\boldsymbol{\theta}_i}$ and estimate the unregularized action-value function:

$$Q_{\boldsymbol{\theta}_i}^q(s, a) = Q_{\boldsymbol{\theta}_i}(s, a) = \mathbb{E}_{\tau_{\pi_i}, s_0=s, a_0=a} \left[ \sum_t^{\infty} \gamma^t r_t \right], \tag{5}$$

since $\mathrm{KL}(q \| \pi_i) = 0$. In practice we estimate $Q_{\boldsymbol{\theta}_i}$ from off-policy data (we refer to Section 4 for details about the policy evaluation step). This greatly increases the data efficiency of our algorithm. Given $Q_{\boldsymbol{\theta}_i}$ we improve the lower bound $\mathcal{J}$ w.r.t. $q$ by first expanding $Q_{\boldsymbol{\theta}_i}(s, a)$ via the regularized Bellman operator $T^{\pi, q} = \mathbb{E}_{q(a|s)} \Big[ r(s, a) - \alpha \mathrm{KL}(q \| \pi_i) + \gamma \mathbb{E}_{p(s'|s, a)}[V_{\boldsymbol{\theta}_i}(s')] \Big]$, and optimize the "one-step" KL regularised objective

$$\max_q \bar{\mathcal{J}}_s(q, \theta_i) = \max_q T^{\pi, q} Q_{\boldsymbol{\theta}_i}(s, a)$$

$$= \max_q \mathbb{E}_{\mu(s)} \Big[ \mathbb{E}_{q(\cdot|s)}[Q_{\boldsymbol{\theta}_i}(s, a)] - \alpha \mathrm{KL}(q \| \pi_i) \Big], \tag{6}$$

since $V_{\boldsymbol{\theta}_i}(s) = \mathbb{E}_{q(a|s)}[Q_{\boldsymbol{\theta}_i}(s,a)]$ and thus $Q_{\boldsymbol{\theta}_i}(s,a) = r(s,a) + \gamma V_{\boldsymbol{\theta}_i}(s)$.

Maximizing Equation (6), thus obtaining $q_i = \arg\max \bar{\mathcal{J}}(q, \theta_i)$, does not fully optimize $\mathcal{J}$ since we treat $Q_{\theta_i}$ as constant with respect to $q$. An intuitive interpretation $q_i$ is that it chooses the soft-optimal action for one step and then resorts to executing policy $\pi$. In the language of the EM algorithm this optimization implements a partial E-step. In practice we also choose $\mu_q$ to be the stationary distribution as given through samples from the replay buffer.

CONSTRAINED E-STEP

The reward and the KL terms are on an arbitray relative scale. This can make it difficult to choose $\alpha$. We therefore replace the soft KL regularization with a hard constraint with parameter $\epsilon$, i.e,

$$\max_q \mathbb{E}_{\mu(s)}\Big[\mathbb{E}_{q(a|s)}\Big[Q_{\theta_i}(s,a)\Big]\Big]$$
$$s.t.\mathbb{E}_{\mu(s)}\Big[\mathrm{KL}(q(a|s), \pi(a|s, \boldsymbol{\theta}_i))\Big] < \epsilon. \tag{7}$$

If we choose to explicitly parameterize $q(a|s)$ – option 1 below – the resulting optimisation is similar to that performed by the recent TRPO algorithm for continuous control (Schulman et al., 2015); only in an off-policy setting. Analogously, the unconstrained objective (6) is similar to the objective used by PPO (Schulman et al., 2017b). We note, however, that the KL is reversed when compared to the KL used by TRPO and PPO.

To implement (7) we need to choose a form for the variational policy $q(a|s)$. Two options arise:

1. We can use a parametric variational distribution $q(a|s, \boldsymbol{\theta}^q)$, with parameters $\boldsymbol{\theta}^q$, and optimise Equation (7) via the likelihood ratio or action-value gradients. This leads to an algorithm similar to TRPO/PPO and an explicit M-step becomes unnecessary (see. Alg. 3).

2. We can choose a non-parametric representation of $q(a|s)$ given by one probability factor per sample. To achieve generalization in state space we then fit a parametric policy in the M-step.

Fitting a parametric policy in the M-step is a supervised learning problem, allowing us to employ various regularization techniques at that point. It also makes it easier to enforce the hard KL constraint.

NON PARAMETRIC VARIATIONAL DISTRIBUTION

In the non-parametric case we can obtain the optimal sample based $q$ distribution – the solution to Equation (7) – in closed form (see the appendix for a full derivation), as,

$$q_i(a|s) \propto \pi(a|s, \boldsymbol{\theta}_i) \exp\Big(\frac{Q_{\theta_i}(s,a)}{\eta^*}\Big), \tag{8}$$

where we can obtain $\eta^*$ by minimising the following convex dual function,

$$g(\eta) = \eta\epsilon + \eta \int \mu(s) \log \int \pi(a|s, \boldsymbol{\theta}_i) \exp\Big(\frac{Q_{\theta_i}(s,a)}{\eta}\Big) da\, ds, \tag{9}$$

after the optimisation of which we can evaluate $q_i(a|s)$ on given samples.

This optimization problem is similar to the one solved by relative entropy policy search (REPS) (Peters et al., 2010a) with the difference that we optimise only for the conditional variational distribution $q(a|s)$ instead of a joint distribution $q(a, s)$ – effectively fixing $\mu_q(s)$ to the stationary distribution given by previously collected experience – and we use the Q function of the old policy to evaluate the integral over $a$. While this might seem unimportant it *is crucial* as it allows us to estimate the integral over actions with multiple samples without additional environment interaction. This greatly reduces the variance of the estimate and allows for fully off-policy learning at the cost of performing only a partial optimization of $\mathcal{J}$ as described above.

### 3.3 M-STEP

Given $q_i$ from the E-step we can optimize the lower bound $\mathcal{J}$ with respect to $\boldsymbol{\theta}$ to obtain an updated policy $\boldsymbol{\theta}_{i+1} = \arg\max_{\boldsymbol{\theta}} \mathcal{J}(q_i, \boldsymbol{\theta})$. Dropping terms independent of $\boldsymbol{\theta}$ this entails solving for the solution of

$$\max_{\boldsymbol{\theta}} \mathcal{J}(q_i, \theta) = \max_{\boldsymbol{\theta}} \mathbb{E}_{\mu_q(s)}\Big[\mathbb{E}_{q(a|s)}\Big[\log \pi(a|s, \boldsymbol{\theta})\Big]\Big] + \log p(\boldsymbol{\theta}), \tag{10}$$

which corresponds to a weighted maximum a-posteriroi estimation (MAP) problem where samples are weighted by the variational distribution from the E-step. Since this is essentially a supervised learning step we can choose any policy representation in combination with any prior for regularisation. In this paper we set $p(\boldsymbol{\theta})$ to a Gaussian prior around the current policy, i.e, $p(\boldsymbol{\theta}) \approx \mathcal{N}\Big(\mu = \boldsymbol{\theta}_i, \Sigma = \frac{F_{\boldsymbol{\theta}_i}}{\lambda}\Big)$, where $\boldsymbol{\theta}_i$ are the parameters of the current policy distribution, $F_{\boldsymbol{\theta}_i}$ is the empirical Fisher information matrix and $\lambda$ is a positive scalar. As shown in the appendix this suggests the following generalized M-step:

$$\max_{\pi} \mathbb{E}_{\mu_q(s)}\Big[\mathbb{E}_{q(a|s)}\Big[\log \pi(a|s, \boldsymbol{\theta})\Big] - \lambda \text{KL}\Big(\pi(a|s, \boldsymbol{\theta}_i), \pi(a|s, \boldsymbol{\theta})\Big)\Big] \tag{11}$$

which can be re-written as the hard constrained version:

$$\max_{\pi} \mathbb{E}_{\mu_q(s)}\Big[\mathbb{E}_{q(a|s)}\Big[\log \pi(a|s, \boldsymbol{\theta})\Big]\Big]$$
$$s.t. \ \mathbb{E}_{\mu_q(s)}\Big[\text{KL}(\pi(a|s, \boldsymbol{\theta}_i), \pi(a|s, \boldsymbol{\theta}))\Big] < \epsilon. \tag{12}$$

This additional constraint minimises the risk of overfitting the samples, i.e. it helps us to obtain a policy that generalises beyond the state-action samples used for the optimisation. In practice we have found the KL constraint in the M step to greatly increase stability of the algorithm. We also note that in the E-step we are using the reverse, mode-seeking, KL while in the M-step we are using the forward, moment-matching, KL which reduces the tendency of the entropy of the parametric policy to collapse. This is in contrast to other RL algorithms that use M-projection without KL constraint to fit a parametric policy (Peters et al., 2010a; Wirth et al., 2016; Chebotar et al., 2016; Montgomery & Levine, 2016). Using KL constraint in M-step has also been shown effective for stochastic search algorithms (Abdolmaleki et al., 2017).

## 4 POLICY EVALUATION

Our method is directly applicable in an off-policy setting. For this, we have to rely on a stable policy evaluation operator to obtain a parametric representation of the Q-function $Q_\theta(s, a)$. We make use of the policy evaluation operator from the Retrace algorithm Munos et al. (2016), which we found to yield stable policy evaluation in practice[2]. Concretely, we fit the Q-function $Q_{\theta_i}(s, a, \phi)$ as represented by a neural network, with parameters $\phi$, by minimising the squared loss:

$$\min_{\phi} L(\phi) = \min_{\phi} \mathbb{E}_{\mu_b(s), b(a|s)}\Big[\big(Q_{\theta_i}(s_t, a_t, \phi) - Q_t^{\text{ret}}\big)^2\Big], \text{ with}$$

$$Q_t^{\text{ret}} = Q_{\phi'}(s_t, a_t) + \sum_{j=t}^{\infty} \gamma^{j-t} \Big(\prod_{k=t+1}^{j} c_k\Big)\Big[r(s_j, a_j) + \mathbb{E}_{\pi(a|s_{j+1})}[Q_{\phi'}(s_{j+1}, a)] - Q_{\phi'}(s_j, a_j)\Big],$$

$$c_k = \min\Big(1, \frac{\pi(a_k|s_k)}{b(a_k|s_k)}\Big),$$

$$\tag{13}$$

---

[2]We note that, despite this empirical finding, Retrace may not be guaranteed to be stable with function approximation (Touati et al., 2017).

where $Q_{\phi'}(s, a)$ denotes the output of a target Q-network, with parameters $\phi'$, that we copy from the current parameters $\phi$ after each M-step. We truncate the infinite sum after $N$ steps by bootstrapping with $Q_{\phi'}$ (rather than considering a $\lambda$ return). Additionally, $b(a|s)$ denotes the probabilities of an arbitrary behaviour policy. In our case we use an experience replay buffer and hence $b$ is given by the action probabilities stored in the buffer; which correspond to the action probabilities at the time of action selection.

## 5 EXPERIMENTS

For our experiments we evaluate our MPO algorithm across a wide range of tasks. Specifically, we start by looking at the continuous control tasks of the DeepMind Control Suite (Tassa et al. (2018), see Figure 1), and then consider the challenging parkour environments recently published in Heess et al. (2017). In both cases we use a Gaussian distribution for the policy whose mean and covariance are parameterized by a neural network (see appendix for details). In addition, we present initial experiments for discrete control using ATARI environments using a categorical policy distribution (whose logits are again parameterized by a neural network) in the appendix.

### 5.1 EVALUATION ON CONTROL SUITE

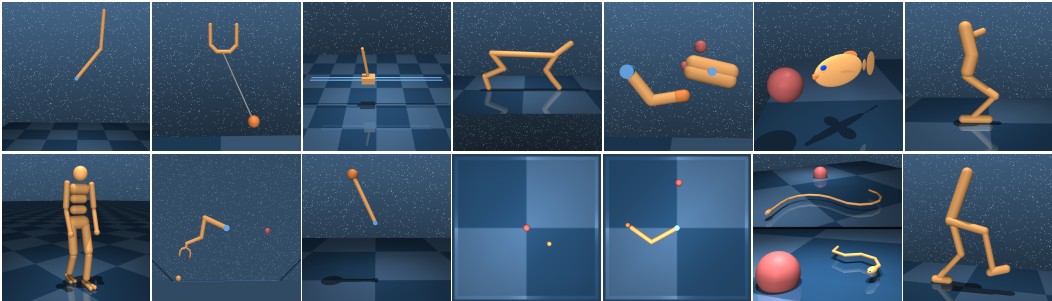

Figure 1: Control Suite domains used for benchmarking. *Top*: Acrobot, Ball-in-cup, Cart-pole, Cheetah, Finger, Fish, Hopper. *Bottom*: Humanoid, Manipulator, Pendulum, Point-mass, Reacher, Swimmers (6 and 15 links), Walker.

The suite of continuous control tasks that we are evaluating against contains 18 tasks, comprising a wide range of domains including well known tasks from the literature. For example, the classical cart-pole and acrobot dynamical systems, 2D and Humanoid walking as well as simple low-dimensional planar reaching and manipulation tasks. This suite of tasks was built in python on top of mujoco and will also be open sourced to the public by the time of publication.

While we include plots depicting the performance of our algorithm on all tasks below; comparing it against the state-of-the-art algorithms in terms of data-efficiency. We want to start by directing the attention of the reader to a more detailed evaluation on three of the harder tasks from the suite.

### 5.1.1 DETAILED ANALYSIS ON WALKER-2D, ACROBOT, HOPPER

We start by looking at the results for the classical Acrobot task (two degrees of freedom, one continuous action dimension) as well as the 2D walker (which has 12 degrees of freedom and thus a 12 dimensional action space and a 21 dimensional state space) and the hopper standing task. The reward in the Acrobot task is the distance of the robots end-effector to an upright position of the underactuated system. For the walker task it is given by the forward velocity, whereas in the hopper the requirement is to stand still.

Figure 2 shows the results for this task obtained by applying our algorithm MPO as well as several ablations – in which different parts were removed from the MPO optimization – and two baselines: our implementation of Proximal Policy Optimization (PPO) (Schulman et al., 2017b) and DDPG. The hyperparameters for MPO were kept fixed for all experiments in the paper (see the appendix for hyperparameter settings).

As a first observation, we can see that MPO gives stable learning on all tasks and, thanks to its fully off-policy implementation, is significantly more sample efficient than the on-policy PPO baseline. Furthermore, we can observe that changing from the non-parametric variational distribution to a parametric distribution[3] (which, as described above, can be related to PPO) results in only a minor asymptotic performance loss but slowed down optimisation and thus hampered sample efficiency; which can be attributed to the fact that the parametric $q$ distribution required a stricter KL constraint. Removing the automatically tuned KL constraint and replacing it with a manually set entropy regulariser then yields an off-policy actor-critic method with Retrace. This policy gradient method still uses the idea of estimating the integral over actions – and thus, for a gradient based optimiser, its likelihood ratio derivative – via multiple action samples (as judged by a Q-Retrace critic). This idea has previously been coined as using the expected policy gradient (EPG) (Ciosek & Whiteson, 2017) and we hence denote the corresponding algorithm with EPG + Retrace, which no-longer follows the intuitions of the MPO perspective. EPG + Retrace performed well when the correct entropy regularisation scale is used. This, however, required task specific tuning (c.f. Figure 4 where this hyperparameter was set to the one that performed best in average across tasks). Finally using only a single sample to estimate the integral (and hence the likelihood ratio gradient) results in an actor-critic variant with Retrace that is the least performant off-policy algorithm in our comparison.

Figure 2: Ablation study of the MPO algorithm and comparison to common baselines from the literature on three domains from the control suite. We plot the median performance over 10 experiments with different random seeds.

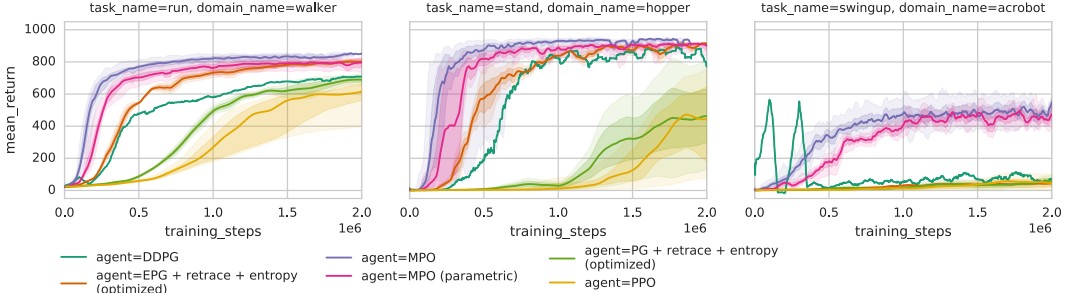

### 5.1.2 COMPLETE RESULTS ON THE CONTROL SUITE

The results for MPO (non-parameteric) – and a comparison to an implementation of state-of-the-art algorithms from the literature in our framework – on all the environments from the control suite that we tested on are shown in Figure 4. All tasks have rewards that are scaled to be between 0 and 1000. We note that in order to ensure a fair comparison all algorithms ran with exactly the same network configuration, used a single learner (no distributed computation), used the same optimizer and were tuned w.r.t. their hyperparameters for best performance across all tasks. We refer to the appendix for a complete description of the hyperparameters. Our comparison is made in terms of data-efficiency.

From the plot a few trends are readily apparent: i) We can clearly observe the advantage in terms of data-efficiency that methods relying on a Q-critic obtain over the PPO baseline. This difference is so extreme that in several instances the PPO baseline converges an order of magnitude slower than the off-policy algorithms and we thus indicate the asymptotic performance of each algorithm of PPO and DDPG (which also improved significantly later during training in some instances) with a colored star in the plot; ii) the difference between the MPO results and the (expected) policy gradient (EPG) with entropy regularisation confirm our suspicion from Section 5.1.1: finding a good setting for the entropy regulariser that transfers across environments without additional constraints on the policy distribution is very difficult, leading to instabilities in the learning curves. In contrast to this the MPO results appear to be stable across all environments; iii) Finally, in terms of data-efficiency the methods utilising Retrace obtain a clear advantage over DDPG. The single learner vanilla DDPG implementation learns the lower dimensional environments quickly but suffers in terms of learning

---

[3]We note that we use a value function baseline $\mathbb{E}_\pi[Q(s, \cdot)]$ in this setup. See appendix for details.

speed in environments with sparse rewards (finger, acrobot) and higher dimensional action spaces. Overall, MPO is able to solve all environments using surprisingly moderate amounts of data. On average less than 1000 trajectories (or $10^6$ samples) are needed to reach the best performance.

## 5.2 HIGH-DIMENSIONAL CONTINUOUS CONTROL

Next we turn to evaluating our algorithm on two higher-dimensional continuous control problems; humanoid and walker. To make computation time bearable in these more complicated domains we utilize a parallel variant of our algorithm: in this implementation K learners are all independently collecting data from an instance of the environment. Updates are performed at the end of each collected trajectory using distributed synchronous gradient descent on a shared set of policy and Q-function parameters (we refer to the appendix for an algorithm description). The results of this experiment are depicted in Figure 3.

For the Humanoid running domain we can observe a similar trend to the experiments from the previous section: MPO quickly finds a stable running policy, outperforming all other algorithms in terms of sample efficiency also in this high-dimensional control problem.

The case for the Walker-2D parkour domain (where we compare against a PPO baseline) is even more striking: where standard PPO requires approximately *1M trajectories* to find a good policy MPO finds a solution that is asymptotically no worse than the PPO solution in **in about 70k trajectories (or 60M samples)**, resulting in an order of magnitude improvement. In addition to the walker experiment we have also evaluated MPO on the Parkour domain using a humanoid body (with 22 degrees of freedom) which was learned successfully (not shown in the plot, please see the supplementary video).

Figure 3: MPO on high-dimensional control problems (Parkour Walker2D and Humanoid walking from control suite).

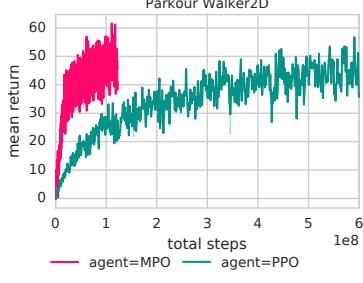
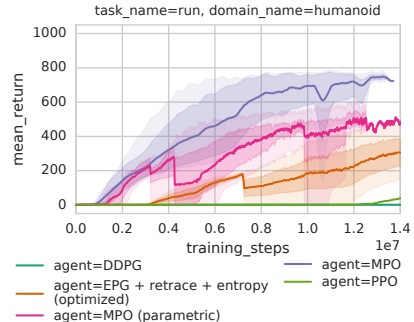

## 5.3 DISCRETE CONTROL

As a proof of concept – showcasing the robustness of our algorithm and its hyperparameters – we performed an experiment on a subset of the games contained contained in the "Arcade Learning Environment" (ALE) where we used *the same hyperparameter* settings for the KL constraints as for the continuous control experiments. The results of this experiment can be found in the Appendix.

## 6 CONCLUSION

We have presented a new off-policy reinforcement learning algorithm called Maximum a-posteriori Policy Optimisation (MPO). The algorithm is motivated by the connection between RL and inference and it consists of an alternating optimisation scheme that has a direct relation to several existing algorithms from the literature. Overall, we arrive at a novel, off-policy algorithm that is highly data efficient, robust to hyperparameter choices and applicable to complex control problems. We demonstrated the effectiveness of MPO on a large set of continuous control problems.

Figure 4: Complete comparison of results for the control suite. We plot the median performance over 10 random seeds together with 5 and 95 % quantiles (shaded area). Note that for DDPG we only plot the median to avoid clutter in the plots. For DDPG and PPO final performance is marked by a star).

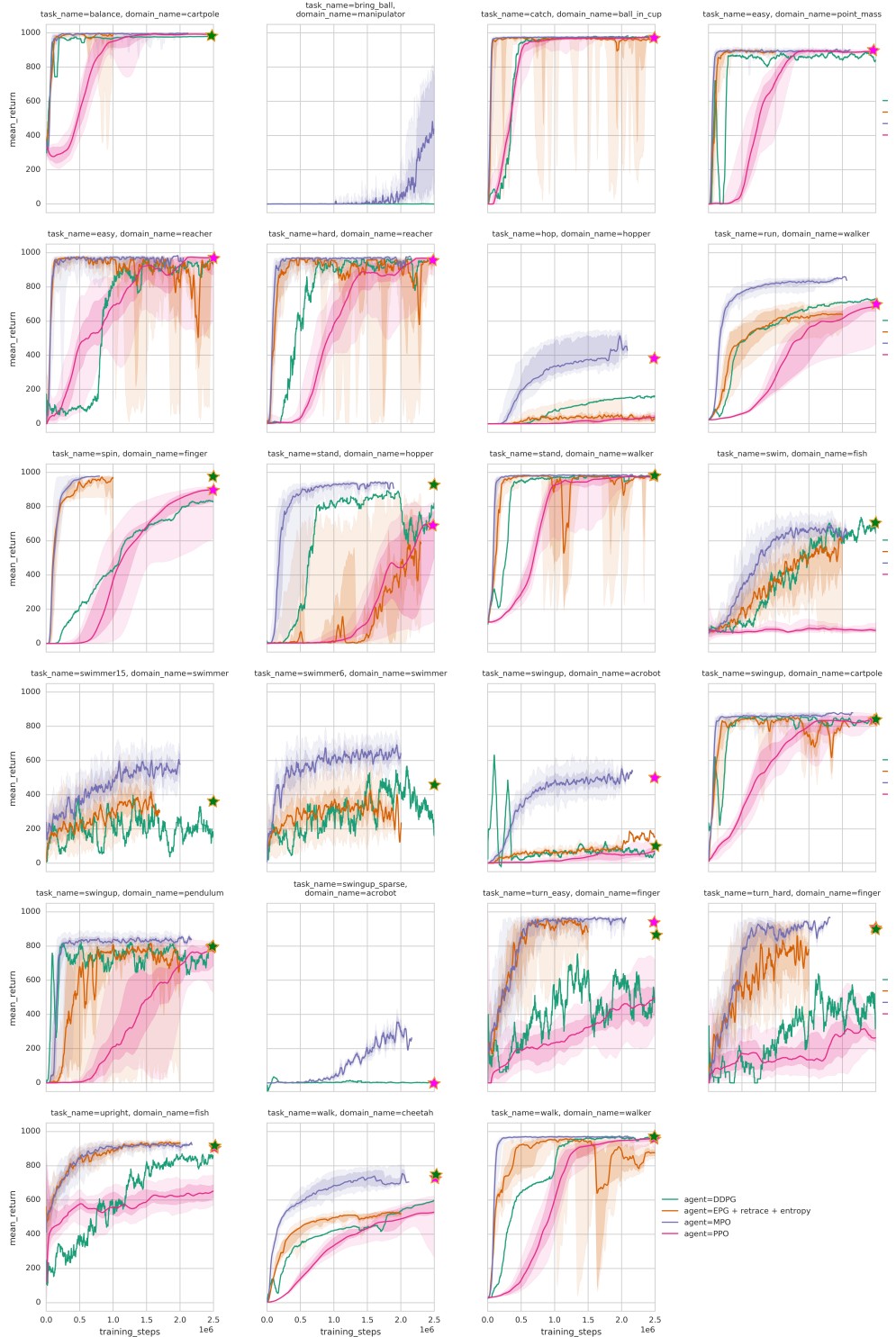

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

# A    PROOF OF MONOTONIC IMPROVEMENT FOR THE KL-REGULARIZED POLICY OPTIMIZATION PROCEDURE

In this section we prove a monotonic improvement guarantee for KL-regularized policy optimization via alternating updates on $\pi$ and $q$ under the assumption that the prior on $\theta$ is uninformative.

## A.1    REGULARIZED REINFORCEMENT LEARNING

Let $\pi$ be an arbitrary policy. For any other policy $q$ such that, for all $x, a$, $\{\pi(a|x) > 0\} \implies \{q(a|x) > 0\}$, define the $\pi$-regularized reward for policy $q$:

$$r_\alpha^{\pi,q}(x,a) = r(x,a) - \alpha \log \frac{q(a|x)}{\pi(a|x)},$$

where $\alpha > 0$.

**Bellman operators:**    Define the $\pi$-regularized Bellman operator for policy $q$

$$T_\alpha^{\pi,q} V(x) = \mathbb{E}_{a \sim q(\cdot|x)} \Big[ r_\alpha^{\pi,q}(x,a) + \gamma \mathbb{E}_{y \sim p(\cdot|x,a)} V(y) \Big],$$

and the non-regularized Bellman operator for policy $q$

$$T^q V(x) = \mathbb{E}_{a \sim q(\cdot|x)} \Big[ r(x,a) + \gamma \mathbb{E}_{y \sim p(\cdot|x,a)} V(y) \Big].$$

**Value function:**    Define the $\pi$-regularized value function for policy $q$ as

$$V_\alpha^{\pi,q}(x) = \mathbb{E}_q \Big[ \sum_{t \geq 0} \gamma^t r_\alpha^{\pi,q}(x_t, a_t) | x_0 = x, q \Big].$$

and the non-regularized value function

$$V^q(x) = \mathbb{E}_q \Big[ \sum_{t \geq 0} \gamma^t r(x_t, a_t) | x_0 = x, q \Big].$$

**Proposition 1.** *For any $q, \pi, V$, we have $V_\alpha^{\pi,q} \leq V^q$ and $T_\alpha^{\pi,q} V \leq T^q V$. Indeed*

$$\mathbb{E}_q \Big[ \log \frac{q(a_t|x_t)}{\pi(a_t|x_t)} \Big] = KL\big(q(\cdot|x_t) \| \pi(\cdot|x_t)\big) \geq 0.$$

**Optimal value function and policy**    Define the optimal regularized value function: $V_\alpha^{\pi,*}(x) = \max_q V_\alpha^{\pi,q}(x)$, and the optimal (non-regularized) value function: $V^*(x) = \max_q V^q(x)$.

The optimal policy of the $\pi$-regularized problem $q_\alpha^{\pi,*}(\cdot|x) = \arg\max_q V_\alpha^{\pi,q}(x)$ and the optimal policy of the non-regularized problem $q^*(\cdot|x) = \arg\max_q V^q$.

**Proposition 2.** *We have that $V_\alpha^{\pi,q}$ is the unique fixed point of $T_\alpha^{\pi,q}$, and $V^q$ is the unique fixed point of $T^q$. Thus we have the following Bellman equations: For all $x \in X$,*

$$V_\alpha^{\pi,q}(x) = \sum_a q(a|x) \Big[ r_\alpha^{\pi,q}(x,a) + \gamma \mathbb{E}_{y \sim p(\cdot|x,a)} \big[ V_\alpha^{\pi,q}(y) \big] \Big] \tag{14}$$

$$V^q(x) = \sum_a q(a|x) \Big[ r(x,a) + \gamma \mathbb{E}_{y \sim p(\cdot|x,a)} \big[ V^q(y) \big] \Big] \tag{15}$$

$$V_\alpha^{\pi,*}(x) = r_\alpha^{\pi,q_\alpha^{\pi,*}}(x,a) + \gamma \mathbb{E}_{y \sim p(\cdot|x,a)} \big[ V_\alpha^{\pi,*}(y) \big] \text{ for all } a \in A, \tag{16}$$

$$V^*(x) = \max_{a \in A} \Big[ r(x,a) + \gamma \mathbb{E}_{y \sim p(\cdot|x,a)} \big[ V^*(y) \big] \Big]. \tag{17}$$

Notice that (16) holds for all actions $a \in A$, and not in expectation w.r.t. $a \sim q(\cdot|x)$ only.

A.2 REGULARIZED JOINT POLICY GRADIENT

We now consider a parametrized policy $\pi_\theta$ and consider maximizing the regularized joint policy optimization problem for a given initial state $x_0$ (this could be a distribution over initial states). Thus we want to find a parameter $\theta$ that (locally) maximizes

$$\mathcal{J}(\theta, q) = V^{\pi_\theta, q}(x_0) = \mathbb{E}_q\Big[\sum_{t \geq 0} \gamma^t\big(r(x_t, a_t) - \alpha \mathrm{KL}(q(\cdot|x_t)\|\pi_\theta(\cdot|x_t)))\big|x_0, q\Big].$$

We start with an initial parameter $\theta_0$ and define a sequence of policies $\pi_i = \pi_{\theta_i}$ parametrized by $\theta_i$, in the following way:

- Given $\theta_i$, define
$$q_i = \arg\max_q T_\alpha^{\pi_{\theta_i}, q} V^{\pi_{\theta_i}},$$

- Define $\theta_{i+1}$ as
$$\theta_{i+1} = \theta_i - \beta \nabla_\theta \mathbb{E}_{\pi_i}\Big[\sum_{t \geq 0} \gamma^t \mathrm{KL}\big(q_k(\cdot|x_t)\|\pi_\theta(\cdot|x_t)\big)_{|\theta=\theta_i}\big|x_0, \pi_i\Big]. \tag{18}$$

**Proposition 3.** *We have the following properties:*

- *The policy $q_i$ satisfies:*
$$q_i(a|x) = \frac{\pi_i(a|x)e^{\frac{1}{\alpha}Q^{\pi_i}(x,a)}}{\mathbb{E}_{b\sim\pi_i(\cdot|x)}\big[e^{\frac{1}{\alpha}Q^{\pi_i}(x,b)}\big]}, \tag{19}$$

  *where $Q^\pi(x, a) = r(x, a) + \gamma \mathbb{E}_{y\sim p(\cdot|x,a)}V^\pi(y)$.*

- *We have*
$$V_\alpha^{\pi_i, q_i} \geq V^{\pi_i}. \tag{20}$$

- *For $\eta$ sufficiently small, we have*
$$\mathcal{J}(\theta_{i+1}, q_{i+1}) \geq \mathcal{J}(\theta_i, q_i) + cg_i, \tag{21}$$

  *where $c$ is a numerical constant, and $g_i$ is the norm of the gradient (minimized by the algorithm):*
$$g_i = \Big\|\nabla_\theta \mathbb{E}_{\pi_i}\Big[\sum_{t \geq 0} \gamma^t KL\big(q_i(\cdot|x_t)\|\pi_\theta(\cdot|x_t)\big)_{|\theta=\theta_i}\big|x_0, \pi_i\Big]\Big\|.$$

  *Thus we build a sequence of policies $(\pi_{\theta_i}, q_i)$ whose values $\mathcal{J}(\theta_i, q_i)$ are non-decreasing thus converge to a local maximum. In addition, the improvement is lower-bounded by a constant times the norm of the gradient, thus the algorithm keeps improving the performance until the gradient vanishes (when we reach the limit of the capacity of our representation).*

*Proof.* We have

$$q_i(\cdot|x) = \arg\max_q \mathbb{E}_{a\sim q(\cdot|x)}\Big[\underbrace{r(x, a) + \gamma\mathbb{E}_{y\sim p(\cdot|x,a)}V^{\pi_i}(y)}_{Q^{\pi_i}(x,a)} - \alpha\log\frac{q(a|x)}{\pi_i(a|x)}\Big],$$

from which we deduce (19). Now, from the definition of $q_i$, we have

$$T_\alpha^{\pi_i, q_i}V^{\pi_i} \geq T_\alpha^{\pi_i, \pi_i}V^{\pi_i} = T^{\pi_i}V^{\pi_i} = V^{\pi_i}.$$

Now, since $T_\alpha^{\pi_i, q_i}$ is a monotone operator (i.e. if $V_1 \geq V_2$ elementwise, then $T_\alpha^{\pi_i, q_i}V_1 \geq T_\alpha^{\pi_i, q_i}V_2$) and its fixed point is $V_\alpha^{\pi_i, q_i}$, we have

$$V_\alpha^{\pi_i, q_i} = \lim_{t\to\infty}(T_\alpha^{\pi_i, q_i})^t V^{\pi_i} \geq V^{\pi_i},$$

which proves (20).

Now, in order to prove (21) we derive the following steps.

**Step 1:** From the definition of $q_{i+1}$ we have, for any $x$,

$$\mathbb{E}_{a\sim q_{i+1}}\big[Q^{\pi_{i+1}}(x,a)\big]-\alpha\mathrm{KL}\big(q_{i+1}(\cdot|x)\|\pi_{i+1}(\cdot|x)\big)\geq \mathbb{E}_{a\sim q_i}\big[Q^{\pi_{i+1}}(x,a)\big]-\alpha\mathrm{KL}\big(q_i(\cdot|x)\|\pi_{i+1}(\cdot|x)\big). \tag{22}$$

Writing the functional that we minimize

$$f(\pi,q,\theta)=\mathbb{E}_\pi\Big[\sum_{t\geq 0}\gamma^t\mathrm{KL}\big(q(\cdot|x_t)\|\pi_\theta(\cdot|x_t)\big)\big|x_0,\pi\Big],$$

the update rule is $\theta_{i+1}=\theta_i-\beta\nabla_\theta f(\pi_i,q_i,\theta_i)$. Thus we have that for sufficiently small $\beta$,

$$f(\pi_i,q_i,\theta_{i+1})\leq f(\pi_i,q_i,\theta_i)-\beta g_i, \tag{23}$$

where $g_i=\frac{1}{2}\|\nabla_\theta f(\pi_i,q_i,\theta_i)\|$.

**Step 2:** Now define $\mathcal{F}$:

$$\begin{aligned}
\mathcal{F}(\pi,q,\theta,\pi') &= \mathbb{E}_\pi\Big[\sum_{t\geq 0}\gamma^t\big(\mathbb{E}_{a\sim q}\big[Q^{\pi'}(x_t,a)\big]-\alpha\mathrm{KL}\big(q(\cdot|x_t)\|\pi_\theta(\cdot|x_t)\big)\big)\big|x_0,\pi\Big]\\
&= \delta_{x_0}(I-\gamma P^\pi)^{-1}T_\alpha^{\pi_\theta,q}V^{\pi'}\\
&= \delta_{x_0}(I-\gamma P^\pi)^{-1}T^q V^{\pi'}-f(\pi,q,\theta),
\end{aligned}$$

where $\delta_{x_0}$ is a Dirac (in the row vector $x_0$), and $P^\pi$ is the transition matrix for policy $\pi$.

From (22) and (23) we deduce that

$$\begin{aligned}
\mathcal{F}(\pi_i,q_{i+1},\theta_{i+1},\pi_{i+1}) &\geq \mathcal{F}(\pi_i,q_i,\theta_{i+1},\pi_{i+1})\\
&\geq \mathcal{F}(\pi_i,q_i,\theta_i,\pi_{i+1})+\beta g_i.
\end{aligned}$$

We deduce

$$\begin{aligned}
&\mathcal{F}(\pi_i,q_{i+1},\theta_{i+1},\pi_i)\\
\geq\ &\mathcal{F}(\pi_i,q_i,\theta_i,\pi_i)+\beta g_i\\
&+\mathcal{F}(\pi_i,q_{i+1},\theta_{i+1},\pi_i)-\mathcal{F}(\pi_i,q_{i+1},\theta_{i+1},\pi_{i+1})+\mathcal{F}(\pi_i,q_i,\theta_i,\pi_{i+1})-\mathcal{F}(\pi_i,q_i,\theta_i,\pi_i)\\
=\ &\mathcal{F}(\pi_i,q_i,\theta_i,\pi_i)+\beta g_i+\\
&\underbrace{\mathbb{E}_{\pi_i}\Big[\sum_{t\geq 0}\gamma^t\big(\mathbb{E}_{a\sim q_{i+1}}\big[Q^{\pi_i}(x_t,a)-Q^{\pi_{i+1}}(x_t,a)\big]-\mathbb{E}_{a\sim q_i}\big[Q^{\pi_i}(x_t,a)-Q^{\pi_{i+1}}(x_t,a)\big]\big)\Big]}_{=O(\beta^2)\text{ since }\pi_i=\pi_{i+1}+O(\beta)\text{ and }q_i=q_{i+1}+O(\beta)}
\end{aligned}$$

This rewrites:

$$\delta_{x_0}(I-\gamma^{\pi_i})^{-1}\big(T_\alpha^{q_{i+1},\pi_{i+1}}V^{\pi_i}-T_\alpha^{q_i,\pi_i}V^{\pi_i}\big)\geq \eta g_i+O(\beta^2). \tag{24}$$

**Step 3:** Now a bit of algebra. For two stochastic matrices $P$ and $P'$, we have

$$\begin{aligned}
&(I-\gamma P)^{-1}\\
=\ &(I-\gamma P')^{-1}+\gamma(I-\gamma P)^{-1}(P-P')(I-\gamma P')^{-1}\\
=\ &(I-\gamma P')^{-1}+\gamma\big[(I-\gamma P')^{-1}+\gamma(I-\gamma P)^{-1}(P-P')(I-\gamma P')^{-1}\big](P-P')(I-\gamma P')^{-1}\\
=\ &(I-\gamma P')^{-1}+\gamma(I-\gamma P')^{-1}(P-P')(I-\gamma P')^{-1}\\
&+\gamma^2(I-\gamma P)^{-1}(P-P')(I-\gamma P')^{-1}(P-P')(I-\gamma P')^{-1}.
\end{aligned}$$

Applying this equality to the transition matrices $P^{\pi_k}$ and $P^{\pi_{k+1}}$ and since $\|P^{\pi_{k+1}}-P^{\pi_k}\|=O(\eta)$, we have:

$$\begin{aligned}
&V_\alpha^{q_{i+1},\pi_{i+1}}\\
=\ &(I-\gamma P^{\pi_i})^{-1}r_\alpha^{q_{i+1},\pi_{i+1}}\\
=\ &(I-\gamma P^{\pi_i})^{-1}r_\alpha^{q_{i+1},\pi_{i+1}}+\gamma(I-\gamma P^{\pi_i})^{-1}(P^{\pi_{i+1}}-P^{\pi_i})(I-\gamma P^{\pi_i})^{-1}r_\alpha^{q_{i+1},\pi_{i+1}}+O(\beta^2)\\
=\ &(I-\gamma P^{\pi_i})^{-1}r_\alpha^{q_i,\pi_i}+(I-\gamma P^{\pi_i})^{-1}(r_\alpha^{q_{i+1},\pi_{i+1}}-r_\alpha^{q_i,\pi_i}+\gamma P^{\pi_{i+1}}-\gamma P^{\pi_i})(I-\gamma P^{\pi_i})^{-1}r_\alpha^{q_i,\pi_i}+O(\beta^2)\\
=\ &V_\alpha^{q_i,\pi_i}+(I-\gamma P^{\pi_i})^{-1}(T_\alpha^{q_{i+1},\pi_{i+1}}V^{\pi_i}-T_\alpha^{q_i,\pi_i}V^{\pi_k})+O(\beta^2).
\end{aligned}$$

Table 1: Results on a subset of the ALE environments in comparison to baselines taken from (Belle-mare et al., 2017)

| Game/Agent | Human | DQN | Prior. Dueling | C51 | MPO |
|---|---|---|---|---|---|
| Pong | 14.6 | 19.5 | 20.9 | 20.9 | 20.9 |
| Breakout | 30.5 | 385.5 | 366.0 | **748** | 360.5 |
| Q*bert | 13,455.0 | 13,117.3 | 18,760.3 | **23,784** | 10,317.0 |
| Tennis | -8.3 | 12.2 | 0.0 | **23.1** | 22.2 |
| Boxing | 12.1 | 88.0 | 98.9 | **97.8** | 82.0 |

Finally, using (24), we deduce that

$$
\begin{aligned}
\mathcal{J}(\theta_{i+1}, q_{i+1}) &= V_\alpha^{q_{i+1}, \pi_{i+1}}(x_0) \\
&= V_\alpha^{q_i, \pi_i}(x_0) + \delta_{x_0}(I - \gamma P^{\pi_i})^{-1}(T_\alpha^{q_{i+1}, \pi_{i+1}} V^{\pi_i} - T_\alpha^{q_i, \pi_i} V^{\pi_i}) + O(\beta^2) \\
&\geq \mathcal{J}(\theta_i, q_i) + \eta g_i + O(\beta^2) \\
&\geq \mathcal{J}(\theta_i, q_i) + \frac{1}{2}\eta g_i,
\end{aligned}
$$

for small enough $\eta$. $\qquad\square$

## B  ADDITIONAL EXPERIMENT: DISCRETE CONTROL

As a proof of concept – showcasing the robustness of our algorithm and its hyperparameters – we performed an experiment on a subset of the games contained contained in the "Arcade Learning Environment" (ALE). For this experiment we used *the same hyperparameter* settings for the KL constraints as for the continuous control experiments as well as the same learning rate and merely altered the network architecture to the standard network structure used by DQN Mnih et al. (2015) – and created a seperate network with the same architecture, but predicting the parameters of the policy distribution. A comparison between our algorithm and well established baselines from the literature, in terms of the mean performance, is listed in Table 1. While we do not obtain state-of-the-art performance in this experiment, the fact that MPO is competitive, out-of-the-box in these domains suggests that combining the ideas presented in this paper with recent advances for RL with discrete actions (Bellemare et al., 2017) could be a fruitful avenue for future work.

## C  EXPERIMENT DETAILS

In this section we give the details on the hyper-parameters used for each experiment. All the continuous control experiments use a feed-forward network except for Parkour-2d were we used the same network architecture as in Heess et al. (2017). Other hyper parameters for MPO with non parametric variational distribution were set as follows,

| Hyperparameter | control suite | humanoid |
|---|---|---|
| Policy net | 100-100 | 200-200 |
| Q function net | 200-200 | 300-300 |
| $\epsilon$ | 0.1 | " |
| $\epsilon_\mu$ | 0.1 | " |
| $\epsilon_\Sigma$ | 0.0001 | " |
| Discount factor ($\gamma$) | 0.99 | " |
| Adam learning rate | 0.0005 | " |

Table 2: Parameters for non-parametric variational distribution

Hyperparameters for MPO with parametric variational distribution were as follows,

| Hyperparameter | control suite tasks | humanoid |
|---|---|---|
| Policy net | 100-100 | 200-200 |
| Q function net | 200-200 | 300-300 |
| $\epsilon_\mu$ | 0.1 | " |
| $\epsilon_\Sigma$ | 0.0001 | " |
| Discount factor ($\gamma$) | 0.99 | " |
| Adam learning rate | 0.0005 | " |

Table 3: Parameters for parametric variational distribution

## D  DERIVATION OF UPDATE RULES FOR A GAUSSIAN POLICY

For continuous control we assume that the policy is given by a Gaussian distribution with a full covariance matrix, i.e, $\pi(\boldsymbol{a}|\boldsymbol{s},\boldsymbol{\theta}) = \mathcal{N}(\mu, \boldsymbol{\Sigma})$. Our neural network outputs the mean $\mu = \mu(s)$ and Cholesky factor $A = A(s)$, such that $\Sigma = AA^T$. The lower triagular factor $A$ has positive diagonal elements enforced by the softplus transform $A_{ii} \leftarrow \log(1 + \exp(A_{ii}))$.

### D.1  NON-PARAMETRIC VARIATIONAL DISTRIBUTION

In this section we provide the derivations and implementation details for the non-parametric variational distribution case for both E-step and M-step.

### D.2  E-STEP

The E-step with a non-parametric variational solves the following program, where we have replaced expectations with integrals to simplify the following derivations:

$$\max_q \int \mu_q(s) \int q(a|s) Q_{\theta_i}(s,a) da ds$$

$$s.t. \int \mu_q(s) \mathrm{KL}(q(a|s), \pi(a|s, \boldsymbol{\theta}_i)) da < \epsilon,$$

$$\iint \mu_q(s) q(a|s) da ds = 1.$$

First we write the Lagrangian equation, i.e,

$$L(q, \eta, \gamma) = \int \mu_q(s) \int q(a|s) Q_{\boldsymbol{\theta}_i}(s,a) da ds +$$
$$\eta \left( \epsilon - \int \mu_q(s) \int q(a|s) \log \frac{q(a|s)}{\pi(a|s, \boldsymbol{\theta}_i)} \right) + \gamma \left( 1 - \iint \mu_q(s) q(a|s) da ds \right).$$

Next we maximise the Lagrangian $L$ w.r.t the primal variable $q$. The derivative w.r.t $q$ reads,

$$\partial q L(q, \eta, \gamma) = Q_{\boldsymbol{\theta}_i}(a,s) - \eta \log q(a|s) + \eta \log \pi(a|s, \boldsymbol{\theta}_i) - (\eta - \gamma).$$

Setting it to zero and rearranging terms we get

$$q(a|s) = \pi(a|s, \boldsymbol{\theta}_i) \exp\left( \frac{Q_{\boldsymbol{\theta}_i}(a,s)}{\eta} \right) \exp\left( -\frac{\eta - \gamma}{\eta} \right).$$

However the last exponential term is a normalisation constant for $q$. Therefore we can write,

$$\exp(-\frac{\eta - \gamma}{\eta}) = \int \pi(a|s, \boldsymbol{\theta}_i) \exp(\frac{Q_{\boldsymbol{\theta}_i}(a,s)}{\eta}) da,$$

$$\gamma = \eta - \eta \log \left( \int \pi(a|s, \boldsymbol{\theta}_i) \exp(\frac{Q_{\boldsymbol{\theta}_i}(a, s)}{\eta}) da \right).$$

Note that we could write $\gamma$ based on $\pi$ and $\eta$. At this point we can derive the dual function,

$$g(\eta) = \eta \epsilon + \eta \int \mu_q(s) \log \left( \int \pi(a|s, \boldsymbol{\theta}_i) \exp(\frac{Q(a, s)}{\eta}) da \right).$$

### D.3 M-STEP

To obtain the KL constraint in the M step we set $p(\theta)$ to a Gaussian prior around the current policy, i.e,

$$p(\boldsymbol{\theta}) \approx \mathcal{N}\left( \mu = \boldsymbol{\theta}_i, \Sigma = \frac{F_{\boldsymbol{\theta}_i}}{\lambda} \right),$$

where $\boldsymbol{\theta}_i$ are the parameters of the current policy distribution, $F_{\boldsymbol{\theta}_i}$ is the empirical Fisher information matrix and $\lambda$.

With this, and dropping constant terms our optimization program becomes

$$\max_\pi \int \mu_q(s) \int q(a|s) \log \pi(a|s, \boldsymbol{\theta}) da ds - \lambda(\boldsymbol{\theta} - \boldsymbol{\theta}_i)^T F_{\boldsymbol{\theta}_i}^{-1}(\boldsymbol{\theta} - \boldsymbol{\theta}_i). \tag{25}$$

We can observe that $(\boldsymbol{\theta} - \boldsymbol{\theta}_i)^T F_{\boldsymbol{\theta}_i}^{-1}(\boldsymbol{\theta} - \boldsymbol{\theta}_i)$ is the second order Taylor approximation of $\int \mu_q(s) \mathrm{KL}(\pi(a|s, \boldsymbol{\theta}_i), \pi(a|s, \boldsymbol{\theta})) ds$ which leads us to the generalized M-step objective:

$$\max_\pi \int \mu_q(s) \int q(a|s) \log \pi(a|s, \boldsymbol{\theta}) da ds - \lambda \int \mu_q(s) \mathrm{KL}(\pi(a|s, \boldsymbol{\theta}_i), \pi(a|s, \boldsymbol{\theta})) ds \tag{26}$$

which corresponds to Equation (11) from the main text, where expectations are replaced by integrals.

After obtaining the non parametric variational distribution in the M step with a Gaussian policy we empirically observed that better results could be achieved by decoupling the KL constraint into two terms such that we can constrain the contribution of the mean and covariance separately i.e.

$$\int \mu_q(s) \mathrm{KL}(\pi_i(a|s, \boldsymbol{\theta}), \pi(a|s, \boldsymbol{\theta})) = C_\mu + C_\Sigma, \tag{27}$$

where

$$C_\mu = \int \mu_q(s) \tfrac{1}{2} (\mathrm{tr}(\Sigma^{-1} \Sigma_i) - n + \ln(\frac{\Sigma}{\Sigma_i})) ds,$$

$$C_\Sigma = \int \mu_q(s) \tfrac{1}{2} (\mu - \mu_i)^T \Sigma^{-1} (\mu - \mu_i) ds.$$

This decoupling allows us to set different $\epsilon$ values for each component, i.e., $\epsilon_\mu, \epsilon_\Sigma$ for the mean, the covariance matrix respectively. Different $\epsilon$ lead to different learning rates. The effectivness of this decoupling has also been shown in Abdolmaleki et al. (2017). We always set a much smaller epsilon for covariance than the mean. The intuition is that while we would like the distribution moves fast in the action space, we also want to keep the exploration to avoid premature convergence.

In order to solve the constrained optimisation in the M-step, we first write the generalised Lagrangian equation, i.e,

$$L(\boldsymbol{\theta}, \eta_\mu, \eta_\Sigma) = \int \mu_q(s) \int q(a|s) \log \pi(a|s, \boldsymbol{\theta}) da ds + \eta_\mu(\epsilon_\mu - C_\mu) + \eta_\Sigma(\epsilon_\Sigma - C_\Sigma)$$

Where $\eta_\mu$ and $\eta_\Sigma$ are Lagrangian multipliers. Following prior work on constraint optimisation, we formulate the following primal problem,

$$\max_{\boldsymbol{\theta}} \min_{\eta_\mu > 0, \eta_\Sigma > 0} L(\boldsymbol{\theta}, \eta_\mu, \eta_\Sigma).$$

In order to solve for $\boldsymbol{\theta}$ we iteratively solve the inner and outer optimisation programs independently: We fix the Lagrangian multipliers to their current value and optimise for $\boldsymbol{\theta}$ (outer maximisation) and then fix the parameters $\boldsymbol{\theta}$ to their current value and optimise for the Lagrangian multipliers (inner minimisation). We continue this procedure until policy parameters $\boldsymbol{\theta}$ and Lagrangian multipliers converge. Please note that the same approach can be employed to bound the KL explicitly instead of decoupling the contribution of mean and covariance matrix to the KL.

### D.4 PARAMETRIC VARIATIONAL DISTRIBUTION

In this case we assume our variational distribution also uses a Gaussian distribution over the action space and use the same structure as our policy $\pi$.

Similar to the non-parametric case for a Gaussian distribution in the M-step we also use a decoupled KL but this time in the E-step for a Gaussian variational distribution. Using the same reasoning as in the previous section we can obtain the following generalized Lagrangian equation:

$$L(\boldsymbol{\theta}^q, \eta_\mu, \eta_\Sigma) = \int \mu_q(s) \int q(a|s; \boldsymbol{\theta}^q) A_i(a, s) dads + \eta_\mu(\epsilon_\mu - C_\mu) + \eta_\Sigma(\epsilon_\Sigma - C_\Sigma).$$

Where $\eta_\mu$ and $\eta_\Sigma$ are Lagrangian multipliers. And where we use the advantage function $A(a, s)$ instead of the Q function $Q(a, s)$, as it empirically gave better performance. Please note that the KL in the E-step is different than the one used in the M-step. Following prior works on constraint optimisation, we can formulate the following primal problem,

$$\max_{\boldsymbol{\theta}^q} \min_{\eta_\mu > 0, \eta_\Sigma > 0} L(\boldsymbol{\theta}^q, \eta_\mu, \eta_\Sigma)$$

In order to solve for $\boldsymbol{\theta}^q$ we iteratively solve the inner and outer optimisation programs independently. In order to that we fix the Lagrangian multipliers to their current value and optimise for $\boldsymbol{\theta}^q$ (outer maximisation), in this case we use the likelihood ratio gradient to compute the gradient w.r.t $\boldsymbol{\theta}^q$. Subsequently we fix the parameters $\boldsymbol{\theta}^q$ to their current value and optimise for Lagrangian multipliers (inner minimisation). We iteratively continue this procedure until the policy parameters $\boldsymbol{\theta}^q$ and the Lagrangian multipliers converges. Please note that the same approach can be used to bound the KL explicitly instead of decoupling the contribution of mean and covariance matrix to the KL. As our policy has the same structure as the parametric variational distribution, the M step in this case reduce to set the policy parameters $\boldsymbol{\theta}$ to the parameters $\boldsymbol{\theta}^q$ we obtained in E-step, i.e,

$$\boldsymbol{\theta}_{i+1} = \boldsymbol{\theta}^q$$

## E IMPLEMENTATION DETAILS

While we ran most of our experiments using a single learner, we implemented a scalable variant of the presented method in which multiple workers collect data independently in an instance of the considered environment, compute gradients and send them to a chief (or parameter server) that performs parameter update by averaging gradients. That is we use distributed synchronous gradient descent. These procedures are described in Algorithms 1 and 2 for the non-parametric case and 3 for the parametric case.

---

**Algorithm 1** MPO (chief)

---

1: Input $G$ number of gradients to average
2: **while** True **do**
3:      initialize N = 0
4:      initialize gradient store $s_\phi = \{\}, s_\eta = \{\}, s_{\eta_\mu} = \{\}, s_{\eta_\Sigma} = \{\} \ s_\theta = \{\}$
5:      **while** $N < G$ **do**
6:          receive next gradient from worker $w$
7:          $s_\phi = s_\phi + [\delta\phi^w]$
8:          $s_\phi = s_\theta + [\delta\theta^w]$
9:          $s_\eta = s_\eta + [\delta\eta^w]$
10:          $s_{\eta_\mu} = s_{\eta_\mu} + [\delta\eta_\mu^w]$
11:          $s_{\eta_\theta} = s_{\eta_\theta} + [\delta\eta_\theta^w]$
12:      update parameters with average gradient from
13:      $s_\phi, s_\eta, s_{\eta_\mu}, s_{\eta_\Sigma} \ s_\theta$
14:      send new parameters to workers

---

**Algorithm 2** MPO (worker) - Non parametric variational distribution

---

1: Input $= \epsilon, \epsilon_\Sigma, \epsilon_\mu, L_{\max}$
2: $i = 0, L_{\text{curr}} = 0$
3: Initialise $Q_{\omega_i}(a, s), \pi(a|s, \boldsymbol{\theta}_i), \eta, \eta_\mu, \eta_\Sigma$
4: **for** each worker **do**
5:      **while** $L_{\text{curr}} > L_{\max}$ **do**
6:          update replay buffer $\mathcal{B}$ with L trajectories from the environment
7:          $k = 0$
8:          // Find better policy by gradient descent
9:          **while** $k < 1000$ **do**
10:              sample a mini-batch $\mathcal{B}$ of $N$ $(s, a, r)$ pairs from replay
11:              sample $M$ additional actions for each state from $\mathcal{B}, \pi(a|s, \boldsymbol{\theta}_i)$ for estimating integrals
12:              compute gradients, estimating integrals using samples
13:              // Q-function gradient:
14:              $\delta_\phi = \partial_\phi L'_\phi(\phi)$
15:              // E-Step gradient:
16:              $\delta\eta = \partial_\eta g(\eta)$
17:              Let: $q(a|s) \propto \pi(a|s, \boldsymbol{\theta}_i) \exp(\frac{Q_{\theta_t}(a, s, \phi')}{\eta})$
18:              // M-Step gradient:
19:              $[\delta_{\eta_\mu}, \delta_{\eta_\Sigma}] = \alpha\partial_{\eta_\mu, \eta_\Sigma} L(\boldsymbol{\theta}_k, \eta_\mu, \eta_\Sigma)$
20:              $\delta_\theta = \partial_{\boldsymbol{\theta}} L(\boldsymbol{\theta}, \eta_{\mu\,k+1}, \eta_{\Sigma\,k+1})$
21:              send gradients to chief worker
22:              wait for gradient update by chief
23:              fetch new parameters $\phi, \theta, \eta, \eta_\mu, \eta_\Sigma$
24:              $k = k + 1$
25:          $i = i + 1, L_{\text{curr}} = L_{\text{curr}} + L$
26:          $\boldsymbol{\theta}_i = \boldsymbol{\theta}, \phi' = \phi$

---

---

**Algorithm 3** MPO (worker) - parametric variational distribution

---

1: Input $= \epsilon_\Sigma, \epsilon_\mu, L_{\max}$
2: $i = 0, L_{\text{curr}} = 0$
3: Initialise $Q_{\omega_i}(a, s), \pi(a|s, \boldsymbol{\theta}_i), \eta, \eta_\mu, \eta_\Sigma$
4: **for** each worker **do**
5:     **while** $L_{\text{curr}} < L_{\max}$ **do**
6:         update replay buffer $\mathcal{B}$ with L trajectories from the environment
7:         $k = 0$
8:         // Find better policy by gradient descent
9:         **while** $k < 1000$ **do**
10:             sample a mini-batch $\mathcal{B}$ of $N$ $(s, a, r)$ pairs from replay
11:             sample $M$ additional actions for each state from $\mathcal{B}$, $\pi(a|s, \boldsymbol{\theta}_k)$ for estimating integrals
12:             compute gradients, estimating integrals using samples
13:             // Q-function gradient:
14:             $\delta_\phi = \partial_\phi L'_\phi(\phi)$
15:             // E-Step gradient:
16:             $[\delta_{\eta_\mu}, \delta_{\eta_\Sigma}] = \alpha \partial_{\eta_\mu, \eta_\Sigma} L(\boldsymbol{\theta}_k, \eta_\mu, \eta_\Sigma)$
17:             $\delta_\theta = \partial_{\boldsymbol{\theta}} L(\boldsymbol{\theta}, \eta_{\mu\,k+1}, \eta_{\Sigma\,k+1})$
18:             // M-Step gradient: In practice there is no M-step in this case as policy and variatinal distribution $q$ use a same structure.
19:             send gradients to chief worker
20:             wait for gradient update by chief
21:             fetch new parameters $\phi, \theta, \eta, \eta_\mu, \eta_\Sigma$
22:             $k = k + 1$
23:     $i = i + 1, L_{\text{curr}} = L_{\text{curr}} + L$
24:     $\boldsymbol{\theta}_i = \boldsymbol{\theta}, \phi' = \phi$

---

