# OpenReview forum: "Maximum a Posteriori Policy Optimisation"
_ICLR.cc/2018/Conference — Accept (Poster)_

### Official Review · AnonReviewer2 · 2017-11-29
**The paper presents an interesting new algorithm for deep reinforcement learning which outperforms state of the art methods.**

**Rating:** 7
**Confidence:** 5

**Review:**

The paper presents a new algorithm for inference-based reinforcement learning for deep RL. The algorithm decomposes the policy update in two steps, an E and an M-step. In the E-step, the algorithm estimates a variational distribution q which is subsequentially used for the M-step to obtain a new policy. Two versions of the algorithm are presented, using a parametric or a non-parametric (sample-based) distribution for q. The algorithm is used in combination with the retrace algorithm to estimate the q-function, which is also needed in the policy update.

This is a well written paper presenting an interesting algorithm. The algorithm is similar to other inference-based RL algorithm, but is the first application of inference based RL to deep reinforcement learning. The results look very promising and define a new state of the art or deep reinforcement learning in continuous control, which is a very active topic right now. Hence, I think the paper should be accepted.


I do have a few comments / corrections / questions about the paper:

- There are several approaches that already use the a combination of the KL-constraint with reverse KL on a non-parametric distribution and subsequently an M-projection to obtain again a parametric distribution, see HiREPS, non-parametric REPS [Hoof2017, JMLR] or AC-REPS [Wirth2016, AAAI]. These algorithms do not use the inference-based view but the trust region justification. As in the non-parametric case, the asymptotic performance guarantees from the EM framework are gone, why is it beneficial to formulate it with EM instead of directly with a trust region of the expected reward?

- It is not clear to me whether the algorithm really optimizes the original maximum a posteriori objective defined in Equation 1. First, alpha changes every iteration of the algorithm while the objective assumes that alpha is constant. This means that we change the objective all the time which is theoretically a bit weird. Moreover, the presented algorithm also changes the prior all the time (in order to introduce the 2nd trust region) in the M-step. Again, this changes the objective, so it is unclear to me what exactly is maximised in the end. Would it not be cleaner to start with the average reward objective (no prior or alpha) and then introduce both trust regions just out of the motivation that we need trust regions in policy search? Then the objective is clearly defined.

- I did not get whether the additional "one-step KL regularisation" is obtained from the lower bound or just added as additional regularisation? Could you explain?

- The algorithm has now 2 KL constraints, for E and M step. Is the epsilon for both the same or can we achieve better performance by using different epsilons?

- I think the following experiments would be very informative:

   - MPO without trust region in M-step

   - MPO without retrace algorithm for getting the Q-value

   - test different epsilons for E and M step

---

> ### Author Response · Authors · 2018-01-05
> **Thank you for your questions and insightful comments.**
>
> We thank you for your questions and insightful comments.
>
> > There are several approaches that already use the a combination of the KL-constraint with reverse KL on a non-
> parametric distribution and subsequently an M-projection to obtain again a parametric distribution, see HiREPS, non-parametric REPS [Hoof2017, JMLR] or AC-REPS [Wirth2016, AAAI].
> > These algorithms do not use the inference-based view but the trust region justification. As in the non-parametric case, the asymptotic performance guarantees from the EM framework are gone, why is it beneficial to formulate it with EM instead of directly with a trust region of the expected reward?
>
> Thank you for pointing out the additional related work. We will include it in the paper. Regarding the EM vs. trust-region question: The benefit of deriving the algorithm from the perspective of an EM-like coordinate ascent is that it motivates and provides a convenient means for theoretical analysis of the two-step procedure used in our approach. See the added a theoretical analysis that was added to the appendix of the paper.
>
> > It is not clear to me whether the algorithm really optimizes the original maximum a posteriori objective defined in Equation 1.
> > First, alpha changes every iteration of the algorithm while the objective assumes that alpha is constant.
> > This means that we change the objective all the time which is theoretically a bit weird.
> > Moreover, the presented algorithm also changes the prior all the time (in order to introduce the 2nd trust region) in the M-step.
> > Again, this changes the objective, so it is unclear to me what exactly is maximised in the end.
> > Would it not be cleaner to start with the average reward objective (no prior or alpha) and then introduce both trust regions
> > just out of the motivation that we need trust regions in policy search? Then the objective is clearly defined.
>
> The reviewers point is well taken. While we think the unconstrained (soft-regularized) is instructive and useful for theoretical analysis the hard-constrained version can indeed be understood as proposed by the reviewer and equally provides important insights. We will  clarify this in the paper and also include an experimental comparison between the soft and hard-regularized cases.
> Regarding your two concerns: For our theoretical guarantee (that we have now derived in the appendix) to hold we have to fix alpha. However, in practice it changes slowly during optimization and converges to a stable value. One can indeed think of the second trust-region as a simple regularizer that prevents overfitting/too large changes in the (sample-based) M-step (similar small changes in the policy are also required by our proof).
>
> - Regarding the additional experiments you asked for:
>
> We agree and have carried out additional experiments that will be included in the final version, preliminary results are as follows:
>
> 1) MPO without trust region in M-step:
> Also works well for low-dimensional problems but is less robust for high-dimensional problems such as the humanoid.
>
> 2) MPO without retrace algorithm for getting the Q-value
> Is significantly slower to reach the same level of performance in the majority of the control suite tasks (retrace + MPO is never worse in any of the control suite tasks).
>
> 3) test different epsilons for E and M step
> The algorithm seems to be robust to settings of epsilon - as long as it is set roughly to the right order of magnitude (10^-3 to 10^-2 for the E-step, 10^-4 to 10^-1 for the M-step). A very small epsilon will, of course, slow down convergence.

---

### Official Review · AnonReviewer3 · 2017-11-30
**Interesting off-policy algorithms with nice results**

**Rating:** 6
**Confidence:** 1

**Review:**

This is an interesting policy-as-inference approach, presented in a reasonably clear and well-motivated way. I have a couple questions which somewhat echo questions of other commenters here. Unfortunately, I am not sufficiently familiar with the relevant recent policy learning literature to judge novelty. However, as best I am aware the empirical results presented here seem quite impressive for off-policy learning.

- When is it possible to normalize the non-parametric q(a|s) in equation (6)? It seems to me this will be challenging in most any situation where the action space is continuous. Is this guaranteed to be Gaussian? If so, I don’t understand why.

– In equations (5) and (10), a KL divergence regularizer is replaced by a “hard” constraint. However, for optimization purposes, in C.3 the hard constraint is then replaced by a soft constraint (with Lagrange multipliers), which depend on values of epsilon. Are these values of epsilon easy to pick in practice? If so, why are they easier to pick than e.g. the lambda value in eq (10)?

---

> ### Author Response · Authors · 2018-01-05
> **We thank the reviewer for comments and thoughtful questions.**
>
> We thank the reviewer for comments and thoughtful questions. We reply to your main concerns in turn below.
>
> > When is it possible to normalize the non-parametric q(a|s) in equation (6)? It seems to me this will be challenging in most any situation where the action space is continuous.
> > Is this guaranteed to be Gaussian? If so, I don’t understand why.
>
> Please see appendix, section C.2. In the parametric case the solution for q(a|s) is trivially normalized when we impose a parametric form that allows analytic evaluation of the normalization function (such as a Gaussian distribution). .
> For the non-parametric case note that the normalizer is given by
> Z(s) = \int \pi_old(a|s) exp( Q(s,a)/eta) da,
> i.e. it is an expectation with respect to our old policy for which we can obtain a MC estimate: \hat{Z}(s) = 1/N \sum_i exp(Q(s,a_i)/eta)    with a_i \sim \pi_old( \cdot | s).
> Thus we can empirically normalize the density for those state-action samples that we use to estimate pi_new in the M-step.
>
> > In equations (5) and (10), a KL divergence regularizer is replaced by a “hard” constraint.
> > However, for optimization purposes, in C.3 the hard constraint is then replaced by a soft constraint (with Lagrange multipliers), which depend on values of epsilon.
> > Are these values of epsilon easy to pick in practice? If so, why are they easier to pick than e.g. the lambda value in eq (10)?
>
> Thank you for pointing out that the reasoning behind this was not entirely easy to follow. We will improve the presentation in the paper. Indeed we found that choosing epsilon can be easier than choosing a multiplier for the KL regularizer. This is due to the fact that the scale of the rewards is unknown a-priori and hence the multiplier that trades of maximizing expected reward and minimizing KL can be expected to change for different RL environments. In contrast to this, when we put a hard constraint on the KL we can explicitly force the policy to stay "epsilon-close" to the last solution - independent of the reward scale. This allows for an easier transfer of hyperparameters across tasks.

---

### Official Review · AnonReviewer1 · 2017-12-05
**some details to discuss**

**Rating:** 5
**Confidence:** 4

**Review:**

This paper studies new off-policy policy optimization algorithm using relative entropy objective and use EM algorithm to solve it. The general idea is not new, aka, formulating the MDP problem as a probabilistic inference problem.

There are some technical questions:
1. For parametric EM case, there is asymptotic convergence guarantee to local optima case; However, for nonparametric EM case, there is no guarantee for that. This is the biggest concern I have for the theoretical justification of the paper.

2. In section 4, it is said that Retrace algorithm from Munos et al. (2016) is used for policy evaluation. This is not true. The Retrace algorithm, is per se, a value iteration algorithm. I think the author could say using the policy evaluation version of Retrace, or use the truncated importance weights technique as used in Retrace algorithm, which is more accurate.

Besides, a minor point: Retrace algorithm is not off-policy stable with function approximation, as shown in several recent papers, such as
“Convergent Tree-Backup and Retrace with Function Approximation”. But this is a minor point if the author doesn’t emphasize too much about off-policy stability.

3. The shifting between the unconstrained multiplier formulation in Eq.9 to the constrained optimization formulation in Eq.10 should be clarified. Usually, an in-depth analysis between the choice of \lambda in multiplier formulation and the \epsilon in the constraint should be discussed, which is necessary for further theoretical analysis.

4. The experimental conclusions are conducted without sound evidence. For example, the author claims the method to be 'highly data efficient' compared with existing approaches, however, there is no strong evidence supporting this claim.


Overall, although the motivation of this paper is interesting, I think there is still a lot of details to improve.

---

> ### Author Response · Authors · 2018-01-05
> **Thank you for your review, we have prepared an additional theoretical analysis and will update the paper**
>
> We appreciate the detailed comments and questions regarding the connection between our method and EM methods. We have addressed your main concern with an additional theoretical analysis of the algorithm, strengthening the paper.
>
> > 1. For parametric EM case, there is asymptotic convergence guarantee to local optima case; However, for nonparametric
> > EM case, there is no guarantee for that. This is the biggest concern I have for the theoretical justification of the paper.
>
> We have derived a proof that gives a monotonic improvement  guarantee for the nonparametric variant of the algorithm under certain circumstances. We will include this proof in the paper. To summarize: Assuming Q can be represented and estimated, the "partial" E-step in combination with an appropriate gradient-based M-step leads to an improvement of the KL regularized objective and guarantees monotonic improvement of the overall procedure under certain circumstances. See also our response to the Anonymous question below.
>
> > 2. In section 4, it is said that Retrace algorithm from Munos et al. (2016) is used for policy evaluation. This is not true.
> > The Retrace algorithm, is per se, a value iteration algorithm. I think the author could say using the policy evaluation version of Retrace,
> > or use the truncated importance weights technique as used in Retrace algorithm, which is more accurate.
>
> We will clarify that we are using the Retrace operator for policy evaluation only (This use case was indeed also analyzed in Munos et al. (2016)).
>
> > Besides, a minor point: Retrace algorithm is not off-policy stable with function approximation, as shown in several recent papers, such as
> > “Convergent Tree-Backup and Retrace with Function Approximation”. But this is a minor point if the author doesn’t emphasize too much about off-policy stability.
>
> We agree that off-policy stability with function approximation is an important open problem that deserves additional attention but not one specific to this method (i.e. any existing DeepRL algorithm shares these concerns). We will add a short note.
>
> > 3. The shifting between the unconstrained multiplier formulation in Eq.9 to the constrained optimization formulation in Eq.10 should be clarified.
> > Usually, an in-depth analysis between the choice of \lambda in multiplier formulation and the \epsilon in the constraint should be discussed, which is necessary for further theoretical analysis.
>
> We now have a detailed analysis of the unconstrained multiplier formulation (see comment above) of our algorithm. In practice we found that implementing updates according to both hard-constraints and using a fixed regularizer worked well for individual domains. Both \lambda and \epsilon can be found via a small hyperparameter search in this case. When applying the algorithm to many different domains  (with widely different reward scales) with the same set of hyperparameters we found it easier to use the hard-constrained version; which is why we placed a focus on it. We will include these experimental results in an updated version of the paper. We believe these observations are in-line with research on hard-constrained/KL-regularized on-policy learning algorithms such as PPO/TRPO (for which explicit connections between the two settings are also ).
>
> > 4. The experimental conclusions are conducted without sound evidence. For example, the author claims the method to be 'highly data efficient' compared with existing approaches, however, there is no strong evidence supporting this claim.
>
> We believe that the large set of experiments we conducted in the experimental section gives evidence for this. Figure 4 e.g. clearly shows the improved data-efficiency MPO gives over our implementations of state-of-the-art RL algorithms for both on-policy (PPO) and off-policy learning (DDPG, policy gradient + Retrace). Further, when looking at the results for the parkour domain we observe an order of magnitude improvement over the reference experiment. We have started additional experiments for parkour with a full humanoid body - leading to similar speedups over PPO - which will be included in the final version and further solidify the claim on a more difficult benchmark.

---

### Public Comment · (anonymous) · 2017-10-31
**Clarification of Equation 1**

These might be very obvious questions, but I failed to derive the last line (line 4) in equation (1) in the paper.

Firstly, I think it would be helpfull to formally define what $$q(\rho)$$ is. My current assumption is:
$$q(\rho) = p(s_0) \prod_1^\infty p(s_{t+1}|a_t, s_t) q(a_t|s_t)$$
where the 'p' distributions are taken to be equal to the real environmental state transitions.

Now, there are a few problems that I encountered when trying to derive equation (1):

1. I think at the end of the line you should have $$+ \log p(\theta)$$ rather than $$+ p(\theta)$$ (I believe this is a typo)

2. In the definition of the log-probabilities, the $$\alpha$$ parameter appears only in the definition of 'p(O=1|\rho)'. The way it appears is as a denominator in the log-probability. In line 4 of equation (1) it has suddenly appeared as a multiplier in front of the log-densities of $$\pi(a|s_t)$$ and $$q(a|s_t)$$. This is possible if we factor out the $$\alpha^{-1}$$ from the sum of the rewards, but then on that line, there should be a prefactor of $$\alpha^{-1}$$ in front of the expectation over 'q' which seems missing. (I believe this is a typo as well).

3. In the resulting expectation, it is a bit unclear how did the discount factors $$\gamma^t$$ have appeared as well as in front of the rewards also in front of the KL divergences? From the context provided I really failed to be able to account for this, and given that for the rest of the paper this form has been used more than once I was wondering if you could provide some clarification on the derivation of the equation as it is not obvious to at least some of the common readers of the paper.

---

> ### Author Response · Authors · 2017-11-28
> **Re: Clarifications**
>
> Thank you for carefully reading of the paper and uncovering a few minor mistakes.
>
> > Firstly, I think it would be helpfull to formally define what $$q(\rho)$$ is.  My current assumption is: $$q(\rho) = p(s_0) \prod_1^\infty p(s_{t+1}|a_t, s_t) q(a_t|s_t)$$.
> Your assumption is correct. q(\rho) is analogous to p(\rho) (as described in the background section on MDPs). We will add this definition.
>
> >1. I think at the end of the line you should have $$+ \log p(\theta)$$ rather than $$+ p(\theta)$$ (I believe this is a typo)
> Correct, this is indeed a typo and will be fixed in the next revision of the paper.
>
> > 2. In the definition of the log-probabilities, the $$\alpha$$ parameter appears only in the definition of 'p(O=1|\rho)'. The way it appears is as a denominator in the log-probability. In line 4 of equation (1) it has suddenly appeared as a multiplier in front of the log-densities of $$\pi(a|s_t)$$ and $$q(a|s_t)$$. This is possible if we factor out the $$\alpha^{-1}$$ from the sum of the rewards, but then on that line, there should be a prefactor of $$\alpha^{-1}$$ in front of the expectation over 'q' which seems missing. (I believe this is a typo as well).
>
> In this step we indeed just multiplied with the (non-zero) \alpha. We presume you meant that alpha is then, however, missing in front of the prior p(\theta) here. You are correct and this will be also fixed in the next revision.
>
> > 3. In the resulting expectation, it is a bit unclear how did the discount factors $$\gamma^t$$ have appeared as well as in front of the rewards also in front of the KL divergences? From the context provided I really failed to be able to account for this, and given that for the rest of the paper this form has been used more than once I was wondering if you could provide some clarification on the derivation of the equation as it is not obvious to at least some of the common readers of the paper.
>
> Thank you for pointing out this inconsistency which has arisen due to some last minute changes in notation that we introduced when we unified the notation in the paper - switching from presenting the finite-horizon, undiscounted, setting to using the infinite-horizon formulation. As pointed out by previous work (e.g. Rawlik et al.)  there is a direct correspondence between learning / inference in an appropriately constructed graphical model (as suggested by the first line of Eq. 1) and the regularized control objective in the finite horizon, undiscounted case. The regularized RL objective still exists in the discounted, infinite horizon case (e.g. Rawlik et al. or see [1] for another construction), but an equivalent graphical model is harder to construct (and is not of the form currently presented in the paper; e.g. see [1]). We will fix this and clarify the relation in the revision
>
> [1] Probabilistic Inference for Solving Discrete and Continuous State Markov Decision Processes, Marc Toussaint, Amos Storkey, ICML 2004

---

### Public Comment · (anonymous) · 2017-10-31
**Comments**

(1) Clarification of Equation 4

The derivation of "one-step KL regularised objective" is unclear to me and this seems to be related to a partial E-step.

Would you explain this part in more detail?

(2) As far as I know, the previous works on variational RL maximize the marginal log-likelihood p(O=1|\theta) (Toussaint (2009) and Rawlik (2012)), whereas you maximizes the unnormalized posterior p(O=1, \theta) with the prior assumption on $\theta$.
I wonder if the prior assumption enhances the performance.

---

> ### Author Response · Authors · 2017-11-28
> **Thank you for spotting some minor inconsistencies**
>
> Thank you for your thorough read of the paper.
>
> > The derivation of "one-step KL regularised objective" is unclear to me and this seems to be related to a partial E-step.
>
> We will clarify the relationship between the one-step objective and Eq. 1 in more detail in a revised version of the paper. We will also include a proof that the the specific "partial" update we use in the E-step leads to an improvement in Eq. (1) and guarantees monotonic improvement of the overall procedure.
>
> In short, the relation between objective (1) and formula (4) is as follows:
> instead of optimizing objective (1) directly in the E-step (which would entail running soft-Q-learning to convergence - e.g. Q-learning with additional KL terms of subsequent time-steps in a trajectory added to the rewards) we start from the "unregularized" Q-function (Eq. (3)) and expand it via the "regularized"  Bellman operator T Q(s,a) = E_a[Q(s,a)] + \alpha KL(q || \pi). We thus only consider the KL at a given state s in the E-step and not the "full" objective from (1). Nonetheless, as mentioned above we have now prepared a proof that this still leads to an improvement in (1).
>
> > (2) As far as I know, the previous works on variational RL maximize the marginal log-likelihood p(O=1|\theta) (Toussaint (2009) and Rawlik (2012)), whereas you maximizes the unnormalized posterior p(O=1, \theta) with the prior assumption on $\theta$. I wonder if the prior assumption enhances the performance.
>
> Correct. The prior p(\theta) allows us to add regularization to the M-step of our procedure (enforcing a trust-region on the policy). We found this to be important when dealing with hihg-dimensional systems like the humanoid where the M-step could otherwise overfit (as the integral over action is only evaluated using 30 samples in our experiments).

---

### Comment · AnonReviewer2 · 2017-11-29
**Few comments...**


I do have a few comments / corrections / questions about the paper:

- There are several approaches that already use the a combination of the KL-constraint with reverse KL on a non-parametric distribution and subsequently an M-projection to obtain again a parametric distribution, see HiREPS, non-parametric REPS [Hoof2017, JMLR] or AC-REPS [Wirth2016, AAAI]. These algorithms do not use the inference-based view but the trust region justification. As in the non-parametric case, the asymptotic performance guarantees from the EM framework are gone, why is it beneficial to formulate it with EM instead of directly with a trust region of the expected reward?

- It is not clear to me whether the algorithm really optimizes the original maximum a posteriori objective defined in Equation 1. First, alpha changes every iteration of the algorithm while the objective assumes that alpha is constant. This means that we change the objective all the time which is theoretically a bit weird. Moreover, the presented algorithm also changes the prior all the time (in order to introduce the 2nd trust region) in the M-step. Again, this changes the objective, so it is unclear to me what exactly is maximised in the end. Would it not be cleaner to start with the average reward objective (no prior or alpha) and then introduce both trust regions just out of the motivation that we need trust regions in policy search? Then the objective is clearly defined.

- I did not get whether the additional "one-step KL regularisation" is obtained from the lower bound or just added as additional regularisation? Could you explain?

- The algorithm has now 2 KL constraints, for E and M step. Is the epsilon for both the same or can we achieve better performance by using different epsilons?

- I think the following experiments would be very informative:

   - MPO without trust region in M-step

   - MPO without retrace algorithm for getting the Q-value

   - test different epsilons for E and M step

---

### Author Response · Authors · 2018-01-12
**Paper updated**

We have updated the paper to address the concerns raised by the reviewers.
In particular we have included:
 - A detailed theoretical analysis of the MPO framework
 - An updated methods section that has a simpler derivation of the algorithm

---

### Public Comment · (anonymous) · 2018-02-14
**Code**

Hi,

Really impressive work. Do you have any plan to release the code?

---

### Public Comment · (anonymous) · 2018-12-13
**Mathematical mistakes in the derivation of the "generalized M-step"**

There are several mathematical issues with the derivation for the generalized M-step, including in the current arxiv version of the paper.

1. If you are doing a Laplace approximation and do a Gaussian prior around the current policy, it is not the covariance that is equal to the Fisher (even if with temperature), but rather the precision. This surprisingly is not really a typo as it carries out to the derivation in D.3 where the authors claim that the quadratic with the **inverse** Fisher is somehow a second order Taylor expansion to the KL, which is clearly not.

2. The authors in the main text are talking about the **empirical** Fisher. Suddenly, we go to D.3 for the actual derivation and ignoring the mistake discussed in 1. they are motivating the new KL term by being a second order Taylor expansion to the KL. This is flawed, as in fact only the true Fisher has such property and the empirical Fisher has nothing to do with the KL divergence.

3. In terms of second order Taylor expansions of the KL it is well known that the Fisher is the second-order derivative tensor of both the forward and the backward KL. Hence, it remains significantly unjustified why one is chosen above the other without any either theoretical or empirical evidence for the choice.

Finally, the fact that the more stable version decouples the KL term for the term of the mean and the covariance bring in to significant questioning where the whole method has any relevance to "maximum a-posterior" optimization, rather than a Trust Region method. An interesting paper indeed.

---

### Decision · Program_Chairs · 2018-01-29
**ICLR 2018 Conference Acceptance Decision**

**Decision:**

Accept (Poster)

**Comment:**

The main idea of policy-as-inference is not new, but it seems to be the first application of this idea to deep RL, and is somewhat well motivated.  The computational details get a bit hairy, but the good experimental results and the inclusion of ablation studies pushes this above the bar.